# The Clinopathological and Prognostic Significance of *SPOCK1* in Gynecological Cancers: A Bioinformatics Based Analysis

**DOI:** 10.3390/biology14020209

**Published:** 2025-02-16

**Authors:** Enes Karaman, Fatih Yay, Durmus Ayan, Ergul Bayram, Sefa Erturk

**Affiliations:** 1Department of Obstetrics and Gynecology, Faculty of Medicine, Nigde Omer Halisdemir University, 51240 Nigde, Turkey; 2Medical Biochemistry, Nigde Omer Halisdemir University Research and Training Hospital, 51100 Nigde, Turkey; 3Department of Medical Biochemistry, Faculty of Medicine, Nigde Omer Halisdemir University, 51240 Nigde, Turkey; 4Department of Biophysics, Faculty of Medicine, Nigde Omer Halisdemir University, 51240 Nigde, Turkey

**Keywords:** bioinformatics analysis, *SPOCK1*, ovarian serous cystadenocarcinoma, cervical squamous cell carcinoma and endocervical adenocarcinoma, uterine corpus endometrial carcinomas

## Abstract

*SPOCK1* is a gene associated with cancer growth and poor survival rates in various cancers, although its significance in gynecological tumors is not fully understood. *SPOCK1* was investigated in ovarian serous cystadenocarcinoma (OV), cervical cancer (CESC), and endometrial cancer (UCEC) utilizing different databases. *SPOCK1* expression was lower in CESC and UCEC than in normal tissues, but there was no discernible difference in OV. *SPOCK1* was found to be linked to age, cancer stage, weight, and menopausal status in particular cancer types. It also demonstrated weak connections with immune cell infiltration in UCEC, specifically in CD8+ and CD4+ T cells. These data imply that *SPOCK1* may be a helpful prognostic marker and therapeutic target in gynecological malignancies.

## 1. Introduction

Cancer incidence and death rates are dramatically growing year after year. The World Health Organization (WHO) estimates that 20 million new cancer cases and 9.7 million cancer deaths will be reported in 2022 [1]. Although the most frequent malignancies are lung, colorectal, and breast, the prevalence of gynecological cancers among women is increasing year after year. The three most frequent gynecological cancers are cervical, endometrial, and ovarian cancer [2]. Low-stage gynecological malignancies can be treated with surgery, radiation, and chemotherapy [3,4,5]. Despite breakthroughs in clinical treatment, the survival rate for high-grade gynecological malignancies remains very low [6]. As a result, there is urgent need for new diagnostic markers and therapeutic target molecules that can effectively improve the prognosis of cancer.

The extracellular matrix (ECM) contains a heparan sulfate/chondroitin sulfate proteoglycan called kazal-like domain proteoglycan 1 (*SPOCK1*/Testican-1), which was first isolated from seminal plasma in 1992 [7]. It has been expressed in the heart, blood, brain, testis, prostate, and cartilage tissues over time [8,9,10].

Along with controlling the ECM’s dynamic balance, *SPOCK1* also controls cellular characteristics linked to cancer, including invasion, metastasis, and apoptosis. [11]. *SPOCK1* mRNA levels are increased in some cancers, including lung, liver, esophageal, stomach, colorectal, breast, prostate, and head and neck cancers [12]. High levels of *SPOCK1* expression are also linked to metastasis, a poorer prognosis, and a lower survival rate.

*SPOCK1* levels are increased in ovarian cancer and decreased with chemotherapy [13]. However, the relationship between gynecological cancers and *SPOCK1* has not been sufficiently investigated yet.

The present study aimed to determine the clinicopathological features and prognostic significance of *SPOCK1* expression and *SPOCK1*-related signaling pathways in ovarian serous cystadenocarcinoma (OV), cervical squamous cell carcinoma and endocervical adenocarcinoma (CESC), and uterine corpus endometrial carcinomas (UCEC) using Gene Expression Profiling Interactive Analysis (GEPIA2), Genotype-Tissue Expression (GTEx), the Cancer Genome Atlas (TCGA), the Kaplan–Meier plotter, and the University of Alabama at Birmingham CANcer data analysis portal (UALCAN) datasets.

## 2. Materials and Methods

### 2.1. SPOCK1 Gene Expression

GEPIA2 (http://gepia2.cancer-pku.cn/#index, accessed on 10 November 2024) was used to examine the difference in *SPOCK1* gene expression between tumor and normal samples in the cohorts with ovarian serous cystadenocarcinoma (OV), cervical squamous cell carcinoma, endocervical adenocarcinoma (CESC), and uterine corpus endometrial carcinoma (UCEC) [14]. The “BoxPlots” and “ExpressionDIY” tabs were used for the analysis, followed by |Log2FC|. Cutoff: the cutoff is 1 0.05 and the *p*-value. The Cancer Genome Atlas (TCGA) and Geno-type-Tissue Expression (GTEx) were used for normal data. The expression variations between cancer stages were also examined using the “Stage Plot” tab. In OC, there were 426 tumors and 88 normal samples; in CESC, there were 306 tumors and 13 normal samples; and in UCEC, there were 174 tumors and 91 normal samples.

### 2.2. UALCAN Analysis

The UALCAN web tool (https://ualcan.path.uab.edu/, accessed on 10 November 2024), which provides data from TCGA, was used to analyze whether there was a change in *SPOCK1* gene expression due to individual stage, age, tumor grade, and p53 mutation status in OV, CESC, and UCEC malignancies. Welch’s T-test determines statistical significance in group comparisons [15].

### 2.3. Kaplan–Meier (KM) Plotter Analysis

Overall survival (OS) and relapse-free survival (RFS) rate analyses were performed in OV (*n* = 374), CESC (*n* = 304), and UCEC (*n* = 543) patients using the pan-cancer RNAseq database available on the Kaplan–Meier (KM) plotter (https://kmplot.com/analysis/, accessed on 11 November 2024) web tool. Patients were divided into two groups according to the auto-select best cutoff [16].

### 2.4. Timer

Tumor infiltration by different immune cells was analyzed on the TIMER (https://cistrome.shinyapps.io/timer/, accessed on 11 November 2024) website [17]. In the analysis of *SPOCK1* gene expression and immune cell infiltration, purity-corrected partial Spearman’s rho and statistical significance values were obtained. In the analysis of the survival relationship, patients were divided into two groups according to the 50% percentile value in the drawing of KM curves. For all analyses, *p* < 0.05 was considered statistically significant.

### 2.5. cBioPortal

For gene alteration analysis, cBioPortal (www.cbioportal.org, accessed on 11 November 2024), a database providing TCGA-based cancer genomic data, was used [18]. A total of 398, 278, and 509 patient samples with mutation and Copy Number Alteration (CNA) data for OV, CESC, and UCEC were examined, respectively. Putative CNAs were obtained from Genomic Identification of Significant Targets in Cancer (GISTIC) [19].

### 2.6. GeneMANIA

The proteins that *SPOCK1* interacts with the most were obtained using the GeneMania database (https://genemania.org/, accessed on 11 November 2024) [20].

### 2.7. The Human Protein Atlas Database

The Human Protein Atlas database (https://www.proteinatlas.org/, accessed on 11 November 2024) was used to validate the expression of *SPOCK1* at the protein level in cancerous and normal tissues [21,22]. In addition, single-cell RNA expressions of *SPOCK1* in ovarian and endometrial tissues were examined in The Human Protein Atlas database (https://www.proteinatlas.org/, accessed on 11 November 2024) [23].

### 2.8. MicroRNA Target Analysis

The purpose is to investigate the target miRNAs of *SPOCK1* in OV, CESC, and UCEC. By identifying miRNAs associated with these genes, the study aimed to uncover regulatory networks that could provide further insights into their roles in OV, CESC, and UCEC pathogenesis and progression. We used miRDB (MicroRNA Target Prediction Database) (https://mirdb.org/, accessed on 8 February 2025) [24] and TargetScan 8.0 (https://www.targetscan.org/vert_80/, accessed on 8 February 2025) to identify and predict the target genes of differentially expressed miRNAs [25,26].

### 2.9. Association of OV-, CESC-, and UCEC-Associated Long Non-Coding RNAs (LncRNAs) with SPOCK1

The LncRNADisease database (http://www.cuilab.cn/lncrnadisease, accessed on 8 February 2025) was utilized to identify long non-coding RNAs linked to OV, CESC, and UCEC. The LncRNADisease database functions as a platform that provides tools to forecast possible novel lncRNA-disease associations in addition to being a collection of empirically confirmed lncRNA-disease association data. Additionally, it gathers information on lncRNA interactions at several levels, including as DNA, proteins, RNAs, and miRNAs [27].

Utilizing the ENCORI database (https://rnasysu.com/encori/, accessed on 8 February 2025), we investigated the connection between *SPOCK1* and long non-coding RNAs. To disentangle pan-cancer networks involving lncRNAs, miR-NAs, pseudogenes, snoRNAs, RNA-binding proteins (RBPs), and all protein-coding genes, the ENCORI Pan-Cancer Analysis Platform was created. This was accomplished by integrating expression profiles from the TCGA experiment, which included about 10,000 RNA-seq and 9900 miRNA-seq samples from 32 different cancer types [28].

## 3. Results

### 3.1. Gene Expression

The *SPOCK1* gene was not significantly different in OV tissues (*p* > 0.05), but it was lower in CESC and UCEC tumor tissues than in normal tissues (*p* < 0.05). In OV, CESC, and UCEC, there was no discernible variation in *SPOCK1* gene expression by stage (Figure 1). *SPOCK1* gene expression according to clinicopathological features.

*SPOCK1* gene expression in OV was significantly higher in the age range of 61–80 years than in the age range of 41–60 years (*p* = 2.610400 × 10^−2^). Regarding the other clinicopathological characteristics of OV, there was no discernible variation in *SPOCK1* expression (Figure 2). *SPOCK1* gene expression in CESC was higher in control tissues than in stage 1 and stage 2 (*p* = 7.702700 × 10^−3^ and *p* = 2.568400 × 10^−2^, respectively), those aged 21–40 years (*p* = 1.567700 × 10^−2^), and those without nodal metastases (*p* = 1.589840 × 10^−2^). Furthermore, higher *SPOCK1* gene expressions were found in stage1 than in stage3 patients (*p* = 3.958800 × 10^−3^) and in normal-weight than in extreme-weight and obese patients (*p* = 4.244500 × 10^−2^ and *p* = 7.636700 × 10^−3^, respectively) (Figure 3).

*SPOCK1* gene expression in UCEC tissues was higher in control tissues than stage 4 patients (*p* = 4.596900 × 10^−3^), obese patients (*p* = 3.675600 × 10^−2^), those aged 61–80 years old (*p* = 3.393900 × 10^−2^), those with serous histology (*p* = 1.302160 × 10^−2^), and those with TP53-Mutant (*p* = 3.959900 × 10^−2^). Additionally, *SPOCK1* gene expression was higher in stage 1 than stage 4 patients (*p* = 4.009000 × 10^−4^), in stage 2 than stage 4 patients (*p* = 2.404200 × 10^−2^), in stage 3 than stage 4 patients (*p* = 2.920400 × 10^−3^), in the 21–40 age group than in the 61–80 and 81–100 age groups (*p* = 1.194490 × 10^−2^ and *p* = 4.832300 × 10^−2^, respectively), in pre-menopause than in post-menopause patients (*p* = 8.037300 × 10^−3^), in extremely obese than in obese patients (*p* = 1.559710 × 10^−2^), and in those with endometrioid histology than in those with serous histology (*p* = 8.164400 × 10^−4^) (Figure 4).

### 3.2. Prognostic Relationship

High *SPOCK1* gene expression in OV was linked to a shorter OS time but not to the RFS rate (*p* = 0.00047 FDR: 10%). The median survival rates for cohorts with low and high expression were 48.27 and 38.5 months, respectively. While *SPOCK1* and OS rate did not significantly correlate, high *SPOCK1* expression in CESC was linked to a shorter RFS time (*p* = 0.016). OS and RFS rates did not significantly correlate with *SPOCK1* gene expression in UCEC (*p* > 0.05) (Figure 5).

### 3.3. Relationship Between Gene Expression, Immune Infiltration, and Prognosis

*SPOCK1* gene expression in OV exhibited marginally negative associations with dendritic cells, neutrophils, and B cells. (Figure 6A). On the other hand, in CESC, weak positive correlations were observed between *SPOCK1* gene expression and macrophage, while weak negative correlations were observed between B cells and neutrophil (Figure 6B). In UCEC, *SPOCK1* gene expression showed a weak positive correlation with CD8+ T cells and a weak negative correlation with CD4+ T cells (Figure 6C). A shorter OS time (*p* = 0.0039) was linked to decreased dendritic cell infiltration in OV. A shorter survival time was linked to low CD4+ T cell infiltration when assessed based on median survival times in CESC. In UCEC, high B cell and CD8+ T cell infiltration was associated with a shorter OS time (*p* = 0.019 and *p* = 0.022, respectively) (Figure 7).

### 3.4. Changes in the SPOCK1 Gene

*SPOCK1* gene changes were observed in six patients in OV: amplification and missense mutation in one patient, amplification in two patients, deep deletion in one patient, and missense mutation in two patients (Figure 8A,B). One of the missense mutations was in the SPARC_Ca_bdg: secreted protein acidic and rich in cysteine Ca binding region (196–304) domain (K201Q), one in the thyroglobulin_1: thyroglobulin type-1 repeat (313–376) domain (S324C), and one outside the domains (G120W) (Figure 8C). In CESC, one missense A173T mutation was observed in the kazal_2: kazal-type serine protease inhibitor domain (139–180) domain (Figure 8D–F). In UCEC, *SPOCK1* gene changes were observed in 13 patients (Figure 8G). SPARC_Ca_bdg: secreted protein acidic and rich in cysteine Ca binding region (196–304) domain with five missenses (D200N, R204Q, E219K, A221V, and G294D) and one truncated (nonsense E268*) and thyroglobulin_1: thyroglobulin type-1 repeat (313–376) domain, 1 truncated nonsense (E316*), seven missense (R319I/K, A332T, P335S, G350D, S351I, and G353W), and eight missense mutations outside the domains (A105T, R110H, V122M, P138S, P184S, A193S, E410K, and E427V) were observed (Figure 8I). The type and frequency of changes according to cancer types in the UCEC cohort are shown in Figure 8H.

### 3.5. Protein–Protein Interactions

*SPOCK1* interacted mostly with the following proteins: general transcription factor IIE subunit 2; SPARC related modular calcium binding 1; SPARC related modular calcium binding 2; cwcv and kazal-like domains proteoglycan 3; cwcv and kazal-like domains proteoglycan 2; neuropilin 2; tumor associated calcium signal transducer 2; epithelial cell adhesion molecule; secreted protein acidic and cysteine rich; glutamate ionotropic receptor AMPA type subunit 4; CD74 molecule; ankyrin 2; lysyl oxidase like 2; limbic system associated membrane protein; multiple EGF like domains 11; TNF alpha induced protein 6; solute carrier family 24 member 2; follistatin like 1; MLLT10 histone lysine methyltransferase DOT1L cofactor; and solute carrier family 25 members 33. The types and degrees of interaction are given in Table 1 and visualized in Figure 9.

**Table 1 biology-14-00209-t001:** Proteins and types of interactions most frequently interacting with *SPOCK1*.

Gene 1	Gene 2	Weight	Network Group
NRP2	*SPOCK1*	0.019701542	Co-expression
MLLT10	*SPOCK1*	0.008268912	Co-expression
GRIA4	*SPOCK1*	0.01669903	Co-expression
ANK2	*SPOCK1*	0.017251523	Co-expression
LSAMP	*SPOCK1*	0.018749185	Co-expression
SLC24A2	*SPOCK1*	0.01464103	Co-expression
TNFAIP6	*SPOCK1*	0.028061341	Co-expression
LOXL2	*SPOCK1*	0.0097463615	Co-expression
TNFAIP6	*SPOCK1*	0.010739085	Co-expression
MLLT10	*SPOCK1*	0.017006958	Co-expression
MLLT10	*SPOCK1*	0.014998745	Co-expression
NRP2	*SPOCK1*	0.013941652	Co-expression
SPARC	*SPOCK1*	0.0029263757	Co-expression
NRP2	*SPOCK1*	0.008485798	Co-expression
SPARC	*SPOCK1*	0.0060609784	Co-expression
LOXL2	*SPOCK1*	0.015608279	Co-expression
FSTL1	*SPOCK1*	0.018673752	Co-expression
NRP2	*SPOCK1*	0.013626368	Co-expression
LOXL2	*SPOCK1*	0.010063365	Co-expression
LOXL2	*SPOCK1*	0.02081454	Co-expression
SPARC	*SPOCK1*	0.00839866	Co-expression
TNFAIP6	*SPOCK1*	0.010096089	Co-expression
FSTL1	*SPOCK1*	0.013175931	Co-expression
GRIA4	*SPOCK1*	0.020897292	Co-localization
ANK2	*SPOCK1*	0.017005134	Co-localization
LSAMP	*SPOCK1*	0.014932096	Co-localization
MEGF11	*SPOCK1*	0.02718836	Co-localization
SLC24A2	*SPOCK1*	0.01623393	Co-localization
SLC25A33	*SPOCK1*	0.026106538	Co-localization
SMOC1	*SPOCK1*	0.00054169004	Genetic Interactions
SMOC2	*SPOCK1*	0.00030524202	Genetic Interactions
NRP2	*SPOCK1*	0.0001834475	Genetic Interactions
GRIA4	*SPOCK1*	0.0002349689	Genetic Interactions
ANK2	*SPOCK1*	0.00021905602	Genetic Interactions
LSAMP	*SPOCK1*	0.0002118026	Genetic Interactions
SLC24A2	*SPOCK1*	0.00027077497	Genetic Interactions
MLLT10	*SPOCK1*	0.0002457919	Genetic Interactions
GTF2E2	*SPOCK1*	0.46410045	Physical Interactions
SMOC1	*SPOCK1*	0.042663254	Shared protein domains
SMOC2	*SPOCK1*	0.042629868	Shared protein domains
SPOCK3	*SPOCK1*	0.039600287	Shared protein domains
SPOCK2	*SPOCK1*	0.039600287	Shared protein domains
TACSTD2	*SPOCK1*	0.04025133	Shared protein domains
EPCAM	*SPOCK1*	0.040251277	Shared protein domains
SPARC	*SPOCK1*	0.027844463	Shared protein domains
CD74	*SPOCK1*	0.040251113	Shared protein domains
FSTL1	*SPOCK1*	0.014670623	Shared protein domains
SMOC1	*SPOCK1*	0.0494928	Shared protein domains
SMOC2	*SPOCK1*	0.0494928	Shared protein domains
SPOCK3	*SPOCK1*	0.0494928	Shared protein domains
SPOCK2	*SPOCK1*	0.0494928	Shared protein domains
TACSTD2	*SPOCK1*	0.03929862	Shared protein domains
EPCAM	*SPOCK1*	0.03929851	Shared protein domains
SPARC	*SPOCK1*	0.03766745	Shared protein domains
CD74	*SPOCK1*	0.0392984	Shared protein domains
FSTL1	*SPOCK1*	0.01589945	Shared protein domains

**Figure 8 biology-14-00209-f008:**
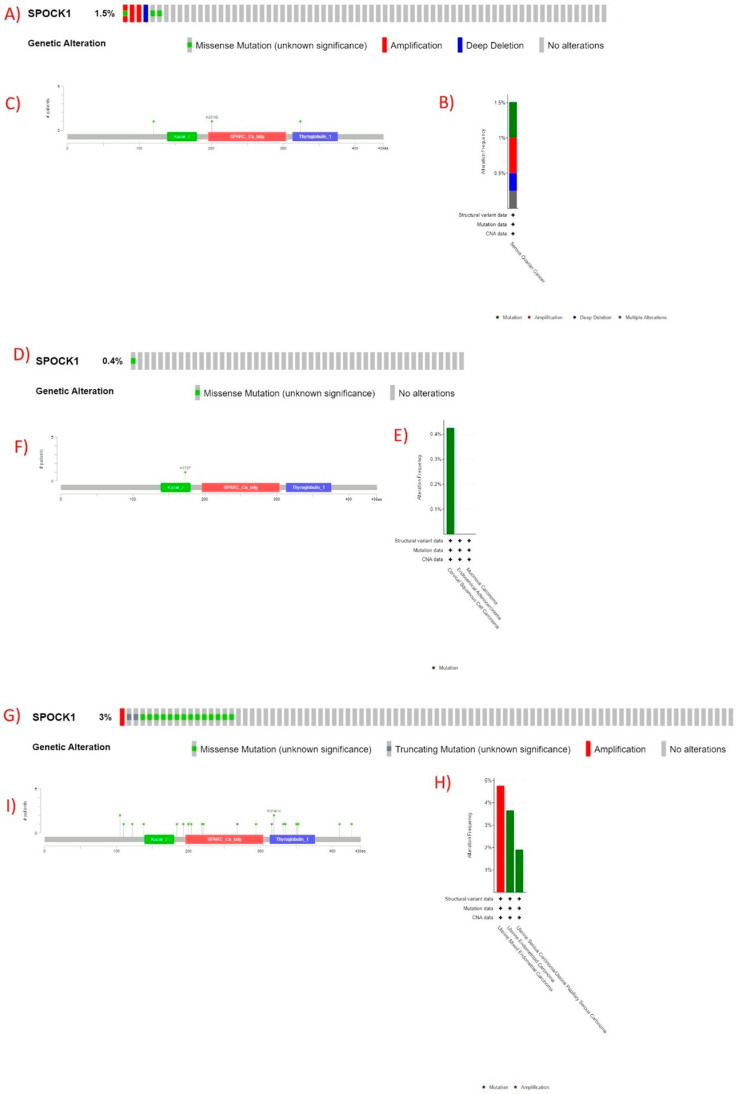
Changes in *SPOCK1* gene in TCGA and pan-cancer atlas datasets. (**A**) Patients with changes in *SPOCK1* in OV cohort, “no alterations” continue until the 398th patient”, (**B**) frequency of changes according to cancer types in OV cohort, (**C**) mutated regions in *SPOCK1* gene in OV cohort, (**D**) patients with changes in *SPOCK1* in CESC cohort, “no alterations” continue until the 278th patient, (**E**) frequency of changes according to cancer types in CESC cohort, (**F**) mutated regions in *SPOCK1* gene in CESC cohort, (**G**) patients with alterations in *SPOCK1* in the UCEC cohort, “no alterations” continue until the 509th patient, (**H**) frequency of alterations according to cancer type in the UCEC cohort, and (**I**) mutated regions in the *SPOCK1* gene in the UCEC cohort. Kazal_2, kazal-type serine protease inhibitor domain (139–180); SPARC_Ca_bdg, secreted protein acidic and rich in cysteine Ca binding region (196–304); and thyroglobulin_1, thyroglobulin type-1 repeat (313–376). Images obtained from the https://www.cbioportal.org/ web tool, accessed on 11 November 2024.

**Figure 9 biology-14-00209-f009:**
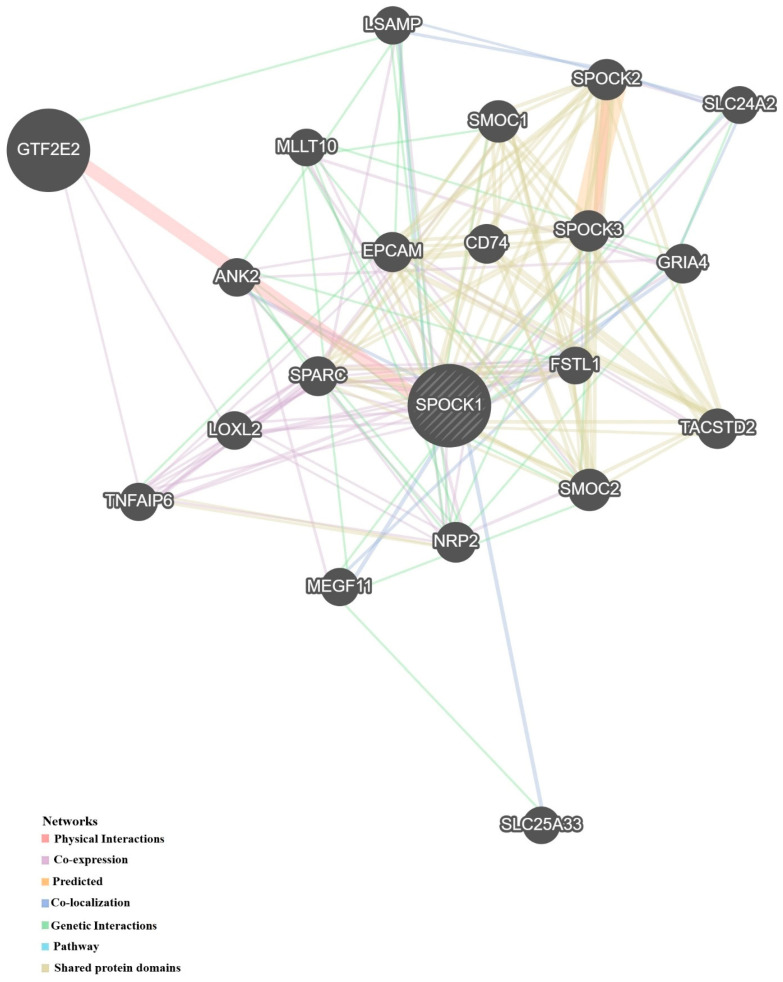
Proteins interacting with *SPOCK1*. The figure was created from GeneMania database (https://genemania.org/, accessed on 11 November 2024). *SPOCK1*, SPARC (osteonectin), cwcv and kazal-like domains proteoglycan 1; *GTF2E2,* general transcription factor IIE subunit 2; *SMOC1*, SPARC related modular calcium binding 1; *SMOC2*, SPARC related modular calcium binding 2; *SPOCK3*, SPARC (osteonectin), cwcv and kazal-like domains proteoglycan 3; *SPOCK2*, SPARC (osteonectin), cwcv, and kazal-like domains proteoglycan 2; *NRP2*, neuropilin 2; *TACSTD2*, tumor associated calcium signal transducer 2; *EPCAM*, epithelial cell adhesion molecule; *SPARC*, secreted protein acidic and cysteine rich; *GRIA4*, glutamate ionotropic receptor AMPA type subunit 4; *CD74*, CD74 molecule; *ANK2*, ankyrin 2; *LOXL2*, lysyl oxidase like 2; *LSAMP*, limbic system associated membrane protein; *MEGF11*, multiple EGF like domains 11; *TNFAIP6*, TNF alpha induced protein 6; *SLC24A2*, solute carrier family 24 member 2; *FSTL1*, follistatin like 1; *MLLT10*, histone lysine methyltransferase DOT1L cofactor; and *SLC25A33*, solute carrier family 25 member 33.

### 3.6. According to the Human Protein Atlas, SPOCK1 Protein

The samples included in the Human Protein Atlas database did not exhibit any expression of the *SPOCK1* protein in normal ovarian tissue. Figure 10 displays a representative sampling of staining properties in additional healthy and malignant tissues.

### 3.7. SPOCK1-Expressing Cells

According to the results in terms of Z score, *SPOCK1* RNA expression in ovarian tissue was in decreasing order: endothelial cells, oocytes, smooth muscle cells, and granulosa cells. In the endometrium, it was seen that it was expressed most in endometrial stromal cells and endothelial cells (Figure 11).

### 3.8. MicroRNA Target Analysis Result

The miRNAs associated with *SPOCK1* obtained from the miRDB database are shown in Figure 12A. Accordingly, miRNAs were grouped as either conserved or poorly conserved. The miRNAs associated with *SPOCK1* obtained from the TargertScan 8.0 database are shown in Figure 12B. Accordingly, miRNAs were grouped as either a high target score or moderate target score. According to both target analyses, the two miRNAs most strongly associated with the *SPOCK1* gene are hsa-miR-19a-3p and hsa-miR-19b-3p.

The effects of hsa-miR-19a-3p and hsa-miR-19b-3p on overall survival (OS) rate in OV, CESC, and UCEC are presented in Figure 13A,B, while their relationship with *SPOCK1* expression is shown in Figure 13C,D. Our analysis revealed that neither hsa-miR-19a-3p nor hsa-miR-19b-3p was significantly associated with OS rates in CESC and UCEC (*p* > 0.05) (Figure 13A,B). On the other hand, we identified a statistically significant negative correlation between *SPOCK1* gene expression and hsa-miR-19a-3p expression in CESC, OV, and UCEC (r = −0.161 and *p* = 4.72 × 10^−3^; r = −0.098 and *p* = 5.74 × 10^−2^; and r = −0.091 and *p* = 3.41 × 10^−2^, respectively) (Figure 13C). However, while a statistically significant negative correlation was observed between *SPOCK1* gene expression and hsa-miR-19b-3p expression in CESC and OV (r = −0.192 and *p* = 7.58 × 10^−4^ and r = −0.115 and *p* = 2.58 × 10^−2^, respectively), no statistically significant correlation was detected in UCEC (r = 0.055 and *p* = 2.00 × 10^−1^) (Figure 13D).

### 3.9. Hepatocellular Cancer-Associated lncRNAs and Their Association with VWF and ADAMST13

The lncRNAs associated with OV, CESC, and UCEC are shown in Figure 14 A–C. We identified HOTAIR as the lncRNA associated with all three cancer types (Figure 14D). Additionally, the common lncRNAs between OV and CESC were identified as BCYRN1, PVT1, and H19, while MALAT1 was found to be shared between CESC and UCEC (Figure 14D).

A significant positive correlation was identified between lncRNA HOTAIR and *SPOCK1* gene expression in the UCEC and OV cohorts (r = 0.116 and *p* = 6.58 × 10^−3^ and r = 0.102 and *p* = 4.68 × 10^−2^, respectively). However, no statistically significant correlation was observed in the CESC cohort (r = 0.011 and *p* = 8.49 × 10^−1^) (Figure 15).

## 4. Discussion

Gynecological cancers are becoming increasingly common in women. Cancers with high malignancy and survival rates of 50–45% include OV, CESC, and UCEC in particular [32,33]. Women who suffer from gynecological cancer have lower survival rates due to the low accuracy of the algorithms and screening concepts employed for early prediction and screening of patients at risk for OV, CESC, and UCEC. As a result, early diagnosis and prognosis greatly depend on the development of predefined treatment targets and prognostic markers including mRNAs, lipids, and proteins [34].

According to recent reports, the *SPOCK1* protein acts as an oncogene, promoting the development and spread of tumors. It has been shown to increase the invasiveness of cancer cells by promoting the epithelial–mesenchymal transition (EMT) and causing metastases in many cancer types. By engaging in the Wnt and PI3K signaling pathways, which are believed to be useful in the development of cancer, researchers have also found that it may make tumor cells more aggressive and malignant. Although this protein’s potential functions in various cancer types have been investigated, gynecological cancer has not been thoroughly studied [35]. Our goal was to use cancer databases to look into the *SPOCK1* protein in three different forms of gynecological cancer (OV, CESC, and UCEC) and analyze its impact as a biomarker for cancer diagnosis and prognosis.

Although *SPOCK1*, a member of the Sparc protein family, has been studied in many types of cancer such as lung [36], prostate [37], breast [38,39], and colorectal [40], the link between *SPOCK1* and gynecological cancers is not well known due to the very few studies conducted on gynecological cancers. Unlike our study, in the in vitro study conducted by Zhang et al. [41] on OV cell lines, it was reported that silencing of *SPOCK1* reduced the proliferation, migration, and invasion of OV cell lines. In the study conducted by Vancza et al. [13], when *SPOCK1* expression was examined in OV cell lines and tissue and serum samples from OV patients, the researchers found that OV tissues and blood samples showed higher levels of *SPOCK1* than healthy controls. Interestingly, *SPOCK1* levels were also noted to be reduced by chemotherapy. The researchers also observed that high *SPOCK1* expression was associated with a shorter OS time. In our study, when *SPOCK1* expression levels were compared in normal and cancerous tissue samples, it was found to be significantly lower, especially in CESC and UCEC tumor tissues (*p* < 0.05). However, no significant difference was observed in OV. In our study, we examined *SPOCK1* gene expression in OV, CESC, and UCEC malignancies depending on individual stage, age, tumor grade, and p53 mutation status. *SPOCK1* gene expression was only associated with age in OVs, while it was associated with age, stage, and weight in CESC and age, stage, histology, weight, and menopausal status in UCEC. In addition, high *SPOCK1* gene expression was associated with shorter OS times in OVs, while no significant relationship existed between high *SPOCK1* gene expression and OS rates in CESC and UCEC. Since our study was based on data obtained from gynecological cancers in the hypothesis databases, the study groups could not be formed in equal numbers. However, we believe that our findings will provide ideas for other studies to be conducted in determining the relationship between gynecological cancers and *SPOCK1* protein. In addition, comprehensive in vivo and in vitro studies including many individuals may create new strategies for cancer diagnosis and treatment by elucidating whether *SPOCK1* can be used as a biomarker.

The *SPOCK1* gene is evolutionarily conserved and contains three fully preserved functional domains: the SPARC/osteonectin CWCV domain, the kazal-like domain, and the thyroglobulin type-1 domain. These domains are associated with functions such as membrane-type matrix metalloproteinase activity, serine protease activity, cell adhesion, and cysteine protease regulation [35].

Particularly, in OV, the p.K201Q, p.S324C, and p.G120W missense mutations are located within splice regions that have been evolutionarily conserved, suggesting that these mutations may lead to functional disruption or significant structural alterations in the protein. Similarly, the p. A173T missense mutation in CESC is located within a functional domain, which may impair protein function. In UCEC, the *SPOCK1* gene harbors five missense mutations (D200N, R204Q, E219K, A221V, and G294D) within the functional domain, along with *two nonsense mutations (E268 and E316*). Notably, these nonsense mutations may result in truncated protein formation, leading to premature termination of translation and potentially generating a completely nonfunctional protein [35].

Three cancers were analyzed for OS and RFS rates, as well as *SPOCK1* expression. As a result, it was found that a shorter OS time was linked to higher *SPOCK1* gene expression in OV. Stated differently, our study found that the expression of the *SPOCK1* gene in OV was negatively correlated with OS rates and reduced the survival rate. Furthermore, a shorter RFS time was linked to higher *SPOCK1* expression in CESC. There was no correlation found between OS and RFS rates and *SPOCK1* expression in UCEC.

The tumor microenvironment (TME) plays an important role in cancer progression [42]. Tumor-infiltrating immune cells are an important part of the TME [43]. Tumor-associated macrophages facilitate tumor invasion by inducing tumor-associated angiogenesis [44]. A meta-analysis has identified a correlation between macrophage infiltration density and poor prognosis in cancer patients [45]. However, tumor-infiltrating CD8+ T cells mediate antitumor immune responses [46]. In our study, we used the TIMER database to reveal the relationship between *SPOCK1* expression and immune cell infiltration. In OV, a weak negative correlation was detected between *SPOCK1* gene expression and B cells, neutrophil, and dendritic cells. In CESC, weak positive correlations were observed between *SPOCK1* gene expression and macrophage, while weak negative correlations were observed between B cells and neutrophils. In UCEC, weak positive correlations were detected with CD8+ T cells, while weak negative correlations were detected with CD4+ T cells. Although our results suggest that *SPOCK1* is associated with the regulation of immune cell infiltration in the TME in gynecological cancers, further research is needed to determine the exact relationship between immune cell infiltration and *SPOCK1* expression in gynecological cancers.

Recent studies have reported a correlation between *SPOCK1* expression and the immune checkpoint molecules PD-1 and PD-L1. The expression of *SPOCK1* showed a positive correlation with PD-1 and PD-L1 [47]. Similarly, in other study, the association between *SPOCK1* expression and PD-L1 expression at the mRNA and protein levels was found to be positive [48].

This suggests that *SPOCK1* could potentially contribute to an immunosuppressive tumor microenvironment by aiding the expression of PD-1/PD-L1. However, the focus of the current study was the association between *SPOCK1* and the infiltration of the lymphoid cells (such as CD4+, CD8+, and dendritic cells) and not the checkpoint inhibition markers. Studies that assess PD-L1 and PD-1 using immunohistochemistry or have transcriptomic analyses should be able to shed some light on this relationship, and we consider this a large space for future work

In the OV, UCEC, and CESC cohorts we analyzed, we identified HOTAIR as a lncRNA associated with all three gynecological cancer types. Studies investigating the relationship between HOTAIR and gynecological cancers have reported that in OV, HOTAIR interacts with PRC2 (Polycomb Repressive Complex 2) to suppress tumor suppressor genes. Moreover, its high expression has been shown to enhance OV cell proliferation and metastatic capacity while promoting tumor progression through the activation of Wnt/β-catenin signaling pathways [49].

Furthermore, HOTAIR has been reported to contribute to the development of paclitaxel resistance, while, in OV, it has been associated with poor prognosis and reduced overall survival rates [50]. In UCEC, HOTAIR has been reported to accelerate the EMT process by reducing E-cadherin levels, thereby enhancing the invasive capacity of tumor cells. Additionally, it has been suggested that HOTAIR promotes metastasis by activating TGF-β signaling pathways while simultaneously suppressing p53 levels, leading to uncontrolled cell proliferation [51]. Since CESC is an HPV-associated cancer, HPV E6/E7 oncoproteins have been reported to enhance HOTAIR expression, thereby inhibiting the pRB and p53 pathways [52]. This phenomenon enables cervical cancer cells to evade apoptosis. Moreover, high HOTAIR expression has been reported to enhance resistance to radiotherapy and chemotherapy agents such as cisplatin, primarily through the activation of the PI3K/Akt and NF-κB signaling pathways [53]. In the OV and UCEC cohorts, we identified a significant positive correlation between HOTAIR gene expression and *SPOCK1* gene expression. The observed positive correlation between HOTAIR and *SPOCK1* suggests that these genes may function synergistically in cancer progression. Additionally, HOTAIR may promote EMT and metastasis by upregulating *SPOCK1* expression, or an inverse relationship could also be possible. Notably, their regulation by common signaling pathways, such as Wnt/β-catenin, PI3K/Akt, or TGF-β signaling pathways, suggests that these two molecules may act in concert in cancer biology. However, it has also been reported that the expression level of HOTAIR is significantly reduced in cisplatin-resistant UCEC cells [54]. At this point, the treatment regimen used may also play a role in the regulation of these genes. hsa-miR-19a-3p and hsa-miR-19b-3p are known as oncogenic microRNAs (oncomiRs) belonging to the miR-17-92 cluster. These microRNAs have been reported to promote cell proliferation, metastasis, epithelial-mesenchymal transition (EMT), and chemotherapy resistance in various cancer types [55,56].

The levels of *SPOCK1* gene expression and hsa-miR-19a-3p expression were found to be negatively correlated in our UCEC, OV, and CESC cohorts. However, only in the CESC and OV cohorts did hsa-miR-19b-3p expression levels exhibit a statistically significant negative correlation with *SPOCK1* gene expression levels. We did not find many research projects examining the connection between these miRNAs and cancer in the literature review. To clarify gynecological cancers, it may be essential to investigate their relationship to cancer, comprehend the nature of the condition, and pay particular attention to how they affect SPOCK gene activity.

This study has demonstrated the relationship between *SPOCK1* protein expression and gynecological cancers. In this study, where protein–protein interactions were also examined, the investigation of possible biomarker roles of *SPOCK1* and its associated proteins with in vivo and in vitro studies may lead to the development of new treatment strategies. The roles of *SPOCK1* in gynecological cancers were examined using data from the TCGA or GEPIA2 database. The findings were derived from a limited cohort with restricted geographic and demographic diversity, potentially compromising generalizability. The substantial reliance on public bioinformatics databases, which may introduce inherent biases, and the absence of comprehensive clinical variables, such as comorbidities and treatment histories, constrain the depth of the analysis. Therefore, the data of patient and control study groups were examined without any criteria. Despite all these limitations, we believe that our findings will be preliminary data for planned human and animal studies.

## 5. Conclusions

It has been determined that *SPOCK1* expression is associated with many clinicopathological features in gynecological cancers such as OV, CSEC, and UCEC. *SPOCK1* may be a potential prognostic and therapeutic target for gynecological cancers. However, this requires comprehensive experimental studies, both genetic and biochemical. Considering that personalized treatment strategies are the main goal of today’s diagnosis and treatment, we believe that it would be beneficial to investigate *SPOCK1*-related proteins. 

## Figures and Tables

**Figure 1 biology-14-00209-f001:**
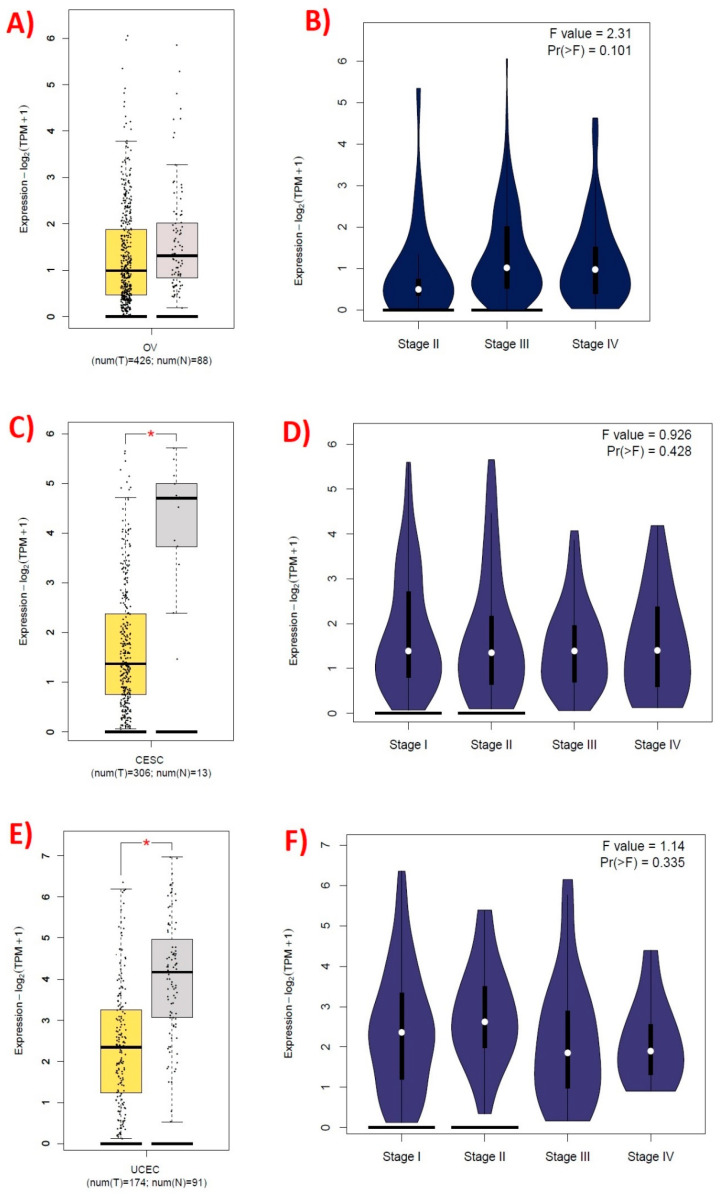
*SPOCK1* gene expression levels in control and tumor tissues. OV, ovarian serous cystadenocarcinoma; CESC, cervical squamous cell carcinoma and endocervical adenocarcinoma; UCEC, uterine corpus endometrial carcinoma; T, tumor; N, control; and TPM, transcripts per million. Created using the GEPIA2 (http://gepia2.cancer-pku.cn/#index, accessed on 11 November 2024) web tool. (**A**) *SPOCK1* gene expression levels of control and tumor tissues in OV; (**B**) *SPOCK1* gene expression levels of tumor tissues in OV stages; (**C**) *SPOCK1* gene expression levels of control and tumor tissues in CESC; (**D**) *SPOCK1* gene expression levels of tumor tissues in CESC stages; (**E**) *SPOCK1* gene expression levels of control and tumor tissues in UCEC; (**F**) *SPOCK1* gene expression levels of tumor tissues in UCEC stages, * *p* < 0.05.

**Figure 2 biology-14-00209-f002:**
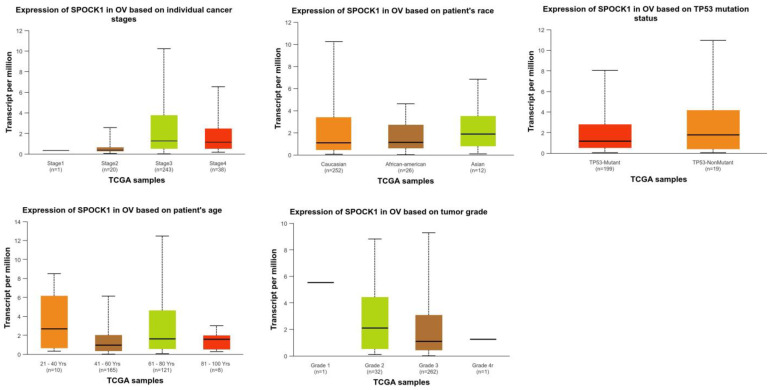
Changes in *SPOCK1* gene expressions according to clinical and pathological features in TCGA samples in the ovarian serous cystadenocarcinoma (OV) cohort. Created using the UALCAN (https://ualcan.path.uab.edu/, accessed on 11 November 2024) web tool.

**Figure 3 biology-14-00209-f003:**
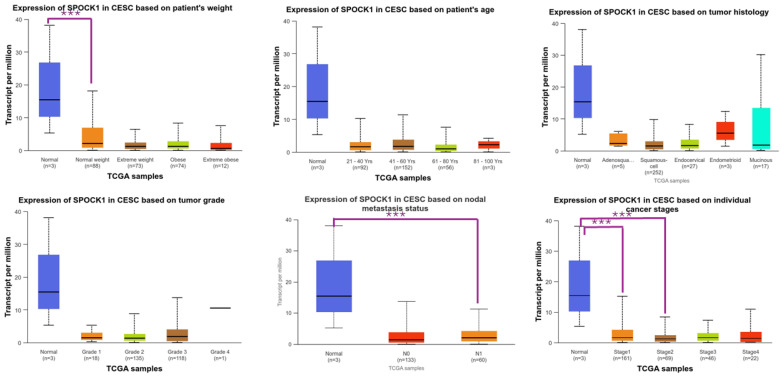
Changes in *SPOCK1* gene expressions according to clinical and pathological features in TCGA samples in the cervical squamous cell carcinoma and endocervical adenocarcinoma (CESC) cohort. Created using the UALCAN (https://ualcan.path.uab.edu/, accessed on 11 November 2024) web tool. *** *p* < 0.05.

**Figure 4 biology-14-00209-f004:**
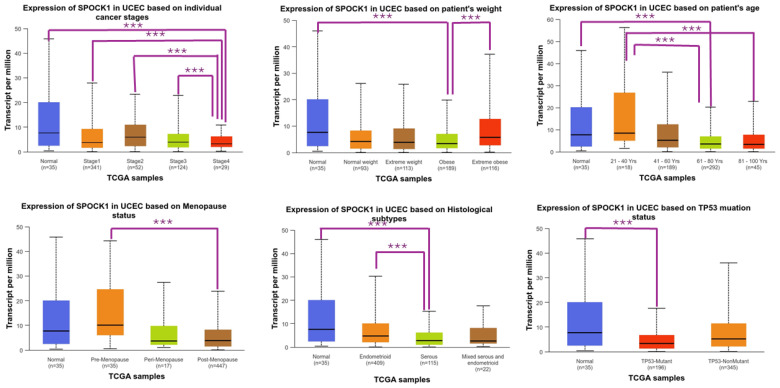
Changes in *SPOCK1* gene expressions according to clinical and pathological features in TCGA samples in the uterine corpus endometrial carcinoma (UCEC) cohort. Created using the UALCAN (https://ualcan.path.uab.edu/, accessed on 11 November 2024) web tool. *** *p* < 0.05.

**Figure 5 biology-14-00209-f005:**
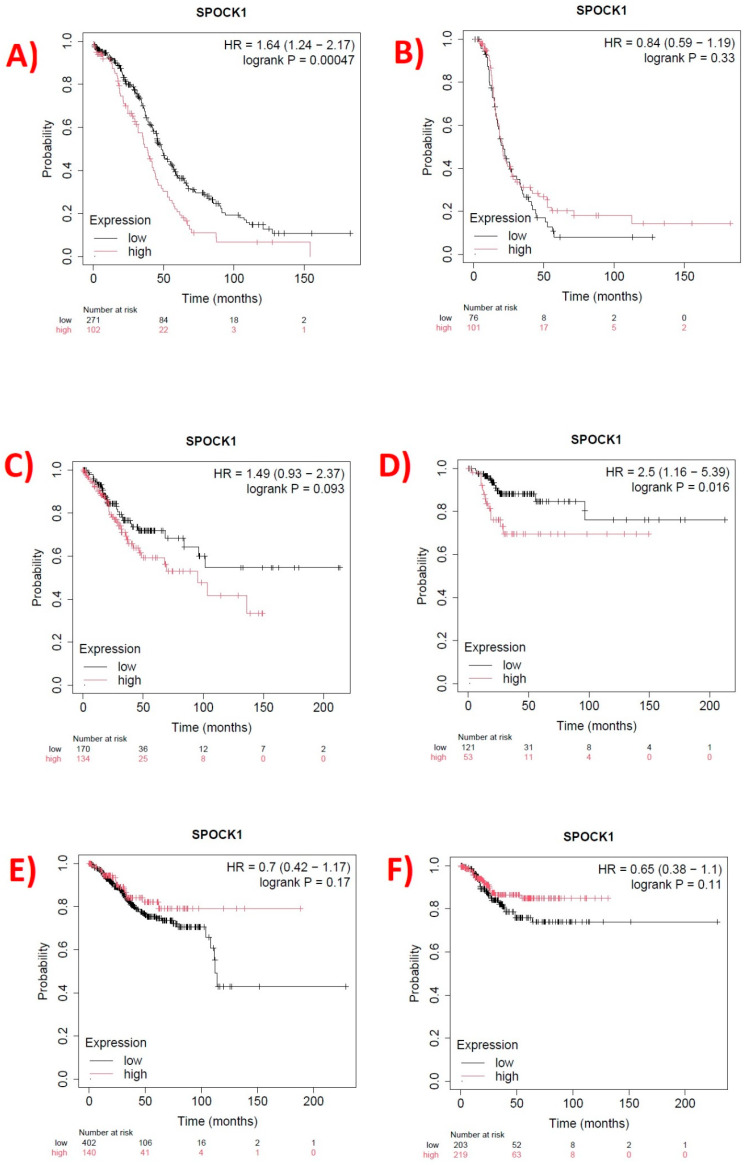
Relationship between *SPOCK1* gene expression and overall survival (OS) and relapse-free survival (RFS) rates. (**A**) OS rate and (**B**) RFS rate in the ovarian cancer cohort, (**C**) OS rate and (**D**) RFS rate in cervical squamous cell carcinoma cohort, and (**E**) OS rate and (**F**) RFS rate in the uterine corpus endometrial carcinoma cohort. Curves were created using https://kmplot.com/analysis/, accessed on 11 November 2024.

**Figure 6 biology-14-00209-f006:**
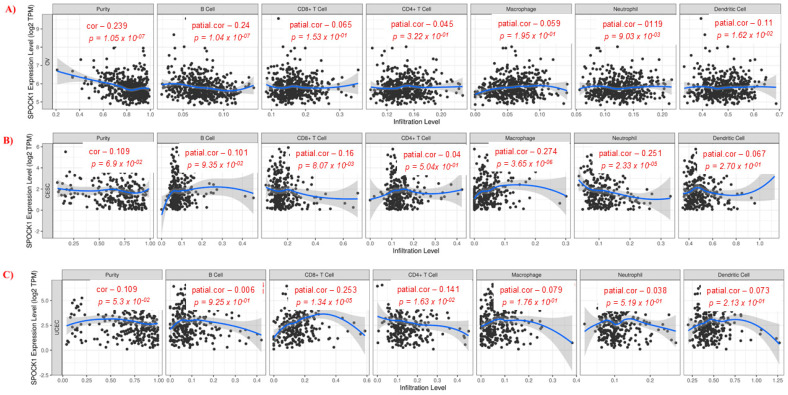
Correlations between immune cell infiltration and *SPOCK1* gene expression. (**A**) Ovarian serous cystadenocarcinoma (OV), (**B**) cervical squamous cell carcinoma and endocervical adenocarcinoma (CESC), and (**C**) uterine corpus endometrial carcinoma (UCEC). https://cistrome.shinyapps.io/timer/ web tool was used, accessed on 11 November 2024.

**Figure 7 biology-14-00209-f007:**
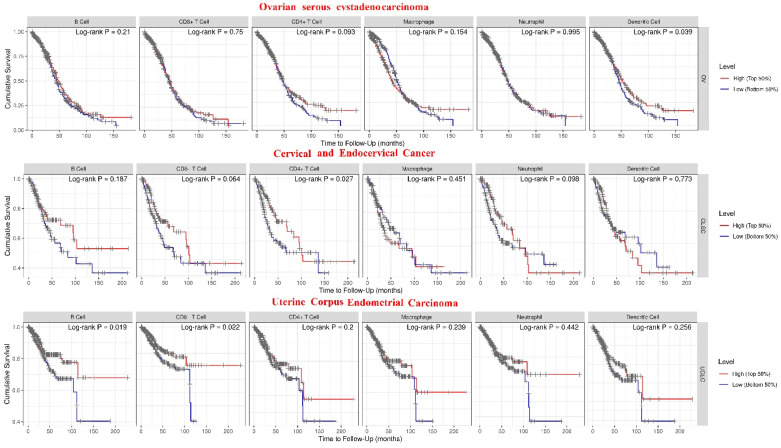
Correlations between immune cell infiltration and *SPOCK1* gene expression. https://cistrome.shinyapps.io/timer/ web tool was used, accessed on 11 November 2024. Immune cell infiltrations and survival associations in ovarian serous cystadenocarcinoma (OV), uterine corpus endometrial carcinoma (UCEC), cervical squamous cell carcinoma and endocervical adenocarcinoma (CESC) cohorts. The same results were obtained with TIMER for OV [29], for UCEC [30], and for CESC [31].

**Figure 10 biology-14-00209-f010:**
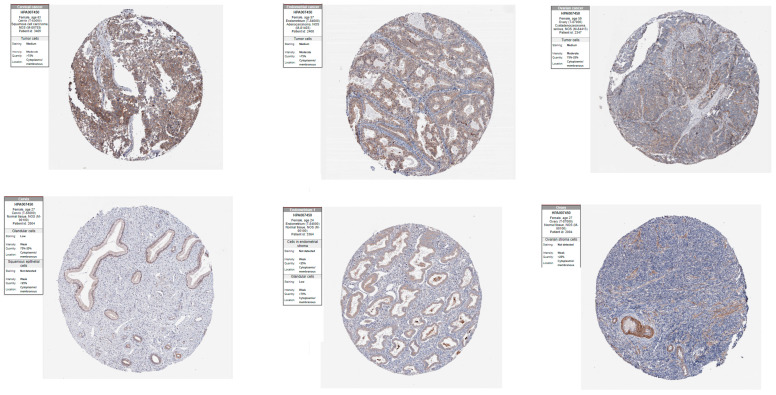
Images of *SPOCK1* staining characteristics in normal and cancer tissue samples of ovary, cervix, and endometrium. URLs: cervical cancer: https://www.proteinatlas.org/ENSG00000152377-SPOCK1/pathology/cervical+cancer#img, accessed on 10 November 2024; normal cervix: https://www.proteinatlas.org/ENSG00000152377-SPOCK1/tissue/Cervix#img, accessed on 10 November 2024; endometrial cancer: https://www.proteinatlas.org/ENSG00000152377-SPOCK1/pathology/endometrial+cancer#img, accessed on 10 November 2024; normal endometrium: https://www.proteinatlas.org/ENSG00000152377-SPOCK1/tissue/Endometrium#img, accessed on 10 November 2024; ovary cancer: https://www.proteinatlas.org/ENSG00000152377-SPOCK1/pathology/ovarian+cancer#img, accessed on 10 November 2024; and normal ovary: no staining was detected in normal ovarian tissues: https://www.proteinatlas.org/ENSG00000152377-SPOCK1/tissue/Ovary#rnaseq, accessed on 10 November 2024.

**Figure 11 biology-14-00209-f011:**
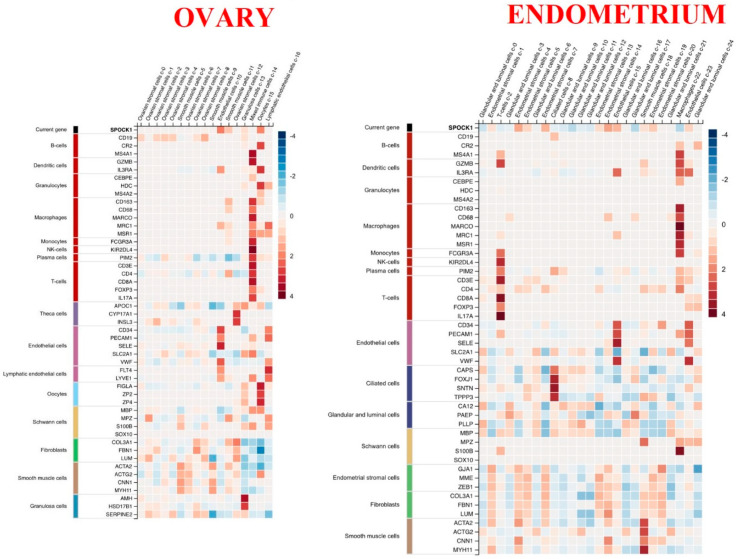
Heat map of gene expressions in single cell types in ovarian and endometrial tissues obtained in terms of Z score. The image is obtained from the Human Protein Atlas. URL for ovary: https://www.proteinatlas.org/ENSG00000152377-SPOCK1/single+cell+type/ovary, accessed on 8 November 2024 and URL for endometrium: https://www.proteinatlas.org/ENSG00000152377-SPOCK1/single+cell+type/endometrium, accessed on 8 November 2024.

**Figure 12 biology-14-00209-f012:**
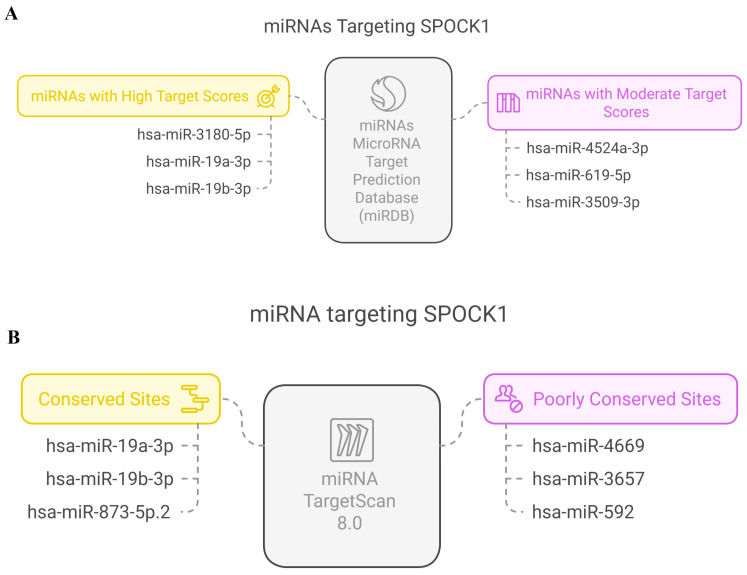
(**A**) The miRNAs associated with *SPOCK1* obtained from miRDB database. (**B**) The miRNAs associated with *SPOCK1* obtained from TargertScan 8.0 database.

**Figure 13 biology-14-00209-f013:**
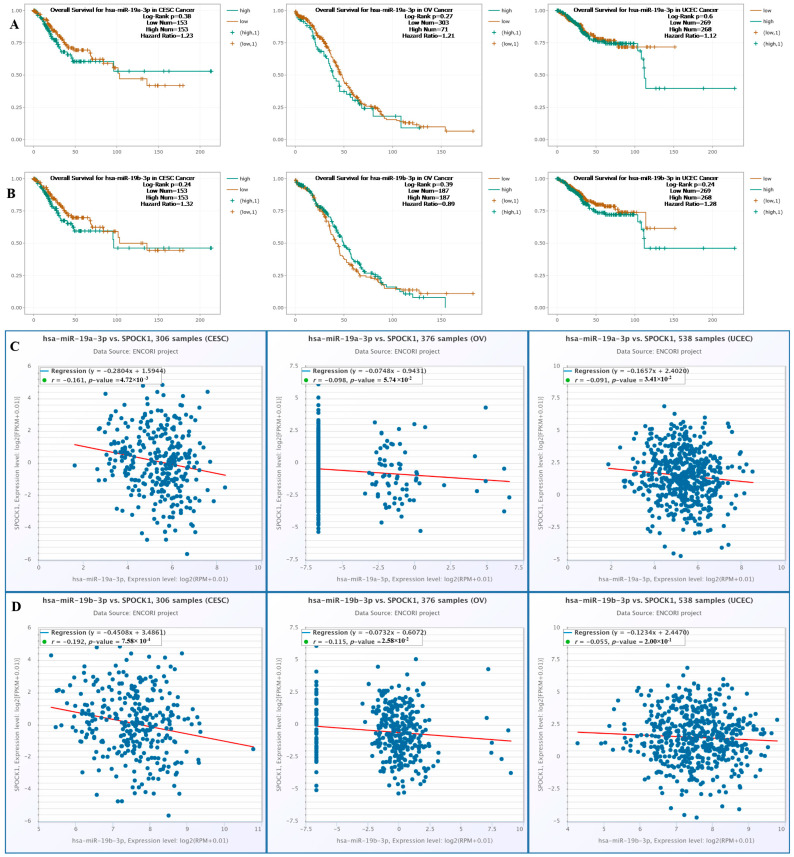
(**A**) Overall survival rates for hsa-miR-19a-3p in CESC, OV, and UCEC. (**B**) Overall survival rates for hsa-miR-19b-3p in CESC, OV, and UCEC. (**C**) Correlation between hsa-miR-19a-3p expression and *SPOCK1* expression in CESC, OV, and UCEC. (**D**) Correlation between hsa-miR-19b-3p expression and *SPOCK1* expression in CESC, OV, and UCEC.

**Figure 14 biology-14-00209-f014:**
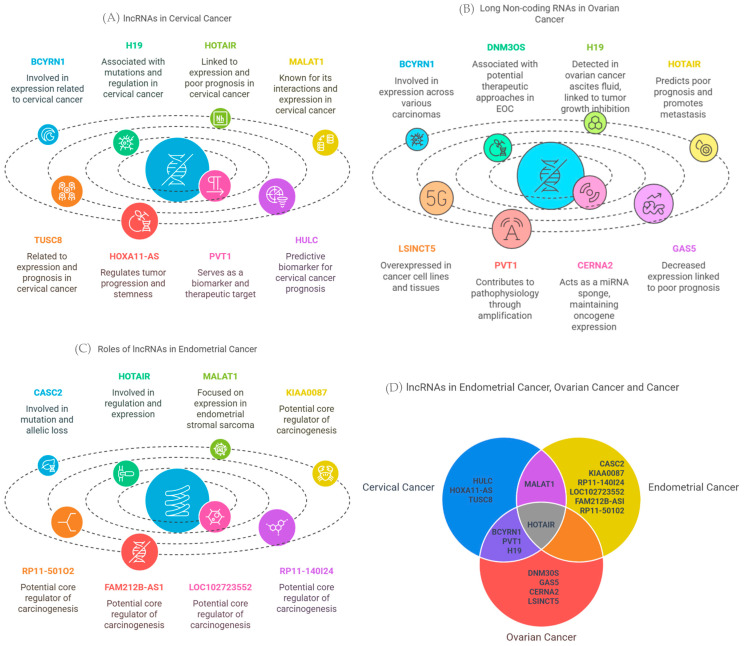
(**A**) LncRNAs associated with CESC, (**B**) LncRNAs associated with OV, (**C**) LncRNAs associated with UCEC, and (**D**) Venn diagram of LncRNAs (from LncRNADisease database (http://www.cuilab.cn/lncrnadisease, accessed on 8 February 2025).

**Figure 15 biology-14-00209-f015:**
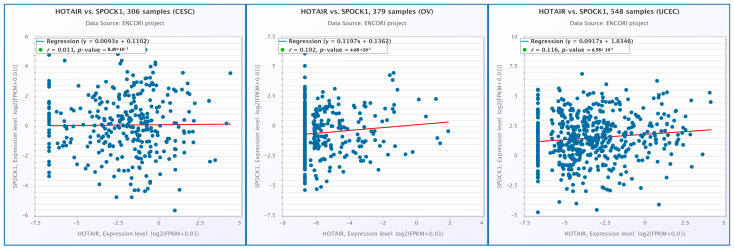
Correlation between HOTAIR and *SPOCK1* expression in CESC, OV, and UCEC.

## Data Availability

The data generated in the present study may be requested from the corresponding author.

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
