# Peer review of "The Clinopathological and Prognostic Significance of SPOCK1 in Gynecological Cancers: A Bioinformatics Based Analysis"

_biology, 2025, doi:10.3390/biology14020209_

Round 1
Reviewer 1 Report
Comments and Suggestions for Authors
I read with great interest the manuscript titled "Clinopathological and Prognostic Significance of SPOCK1 in Gynecological Cancers" by Karaman et al.
The study is both engaging and well-written. However, I have the following observations:
a) The title should specify that the study is based on a bioinformatic analysis.
b) The analysis conducted is exploratory in nature, and this should be acknowledged and discussed in the manuscript.
c) The manuscript should also address the limitations associated with using TCGA datasets and bioinformatic tools.
Comments on the Quality of English LanguageThe manuscript is well-written, with only a few minor grammatical and orthographic errors throughout the text.
Author Response
Commend 1: The title should specify that the study is based on a bioinformatic analysis.
Respond 1: Thank you pointing this out. The title was revised as you suggested.
Commend2: The analysis conducted is exploratory in nature, and this should be acknowledged and discussed in the manuscript.
Respond 2: We appreciate your constructive feedback. It is acknowledged that the analysis conducted in this study is exploratory in nature. This approach was selected to identify potential patterns and associations that could inform future research endeavors. In the revised manuscript, we will explicitly state the exploratory nature of the analysis in the discussion sections. Additionally, we will emphasize that the findings should be interpreted with caution, as they may not establish definitive causal relationships. Furthermore, we will underscore the necessity for confirmatory studies with larger, more representative samples to validate our results.
Commend 3: The manuscript should also address the limitations associated with using TCGA datasets and bioinformatic tools.
Respond 3: Thanks for your valuable contribution. The section was added just before conclusion section.
Reviewer 2 Report
Comments and Suggestions for Authors
In their manuscript "Clinopathological and prognostic significance of SPOCK1 in gynecological cancers", Karaman et al describe their findings on SPOCK1 in gynecological tumors that have been little addressed to date, as well as the associated clinical characterization.
The manuscript is very comprehensive and includes a complete histopathological analysis of approximately 1300 biosamples as well as the corresponding genetic analysis and clinical data processing using the Kaplan-Meier curves described.
The manuscript is well written and has a logical structure. Unfortunately, the figures are often very difficult to decipher due to their small size. In Figure 3 and Figure 4, it would be helpful to indicate significance (e.g. with an asterisk), if significance was achieved.
The authors describe the negative correlation between lymphoid cells (CD4 and CD8), as well as dendritic cells and SPOCK1 expression in relation to a reduced overall survival. At this point, PDl1 and PD1 expression or immunohistochemistry would also have been nice. Is there a correlation here?
The selection of the 3 gynecological cancers ovarian carcinoma, cervical carcinoma and endometrial carcinoma is a good consideration, but from a clinical point of view they have little in common (squamous cell carcinoma (CESC) versus adenocarcinoma (UCEC) or (OC)). The pathophysiology and clinical behavior in terms of initial manifestation, genesis and metastatic behavior are also completely different. It is therefore clinically difficult to draw conclusions from one entity to the other two, but understandable from a biological point of view in view of the global approach. With regard to an evaluation based on a Kaplan-Meier observation, however, the classification is very difficult. Further findings should definitely be awaited here.
Minor layout improvement: In line 28, the word and is missing as a link within cervical squamous cell carcinoma endocervical adenocarcinoma.
Author Response
Commend 1: The manuscript is well written and has a logical structure. Unfortunately, the figures are often very difficult to decipher due to their small size. In Figure 3 and Figure 4, it would be helpful to indicate significance (e.g. with an asterisk), if significance was achieved.
Respond 1: Thank you for your feedback. The figures have been resized for better clarity, and significance has been indicated with asterisks in Figures 3 and 4 where applicable.
Commend 2: The authors describe the negative correlation between lymphoid cells (CD4 and CD8), as well as dendritic cells and SPOCK1 expression in relation to a reduced overall survival. At this point, PDl1 and PD1 expression or immunohistochemistry would also have been nice. Is there a correlation here?
Respond 2: We appreciate the reviewer’s insight into the potential connection between PD-1/PD-L1 expression and SPOCK1. Recent studies have reported a correlation between SPOCK1 expression and the immune checkpoint molecules PD-1 and PD-L1. For instance, one study conducted in BMC Gastroenterology found that expression of SPOCK1 showed a positive correlation with PD-1 and PD-L1 (BMC Gastroenterol, 2022). Similarly, in Frontiers in Cell and Developmental Biology, the association between SPOCK1 expression and PD-L1 expression at the mRNA and protein levels was found to be positive (Front Cell Dev Biol, 2021).
This suggests that SPOCK1 could potentially contribute to an immunosuppressive tumor microenvironment by aiding the expression of PD-1/PD-L1. However, the focus of the current study was the association between SPOCK1 and the infiltration of the lymphoid cells (such as CD4+, CD8+, and dendritic cells) and not the checkpoint inhibition markers. Studies that assess PD-L1 and PD-1 using immunohistochemistry or have transcriptomic analyses should be able to shed some light on this relationship, and we consider this a large space for future work, as we have pointed out in the discussion in the revised manuscript.
To address this point, we have incorporated a discussion on the potential interplay between SPOCK1 and immune checkpoint markers, citing the relevant literature. We sincerely appreciate your valuable suggestions, which have helped enhance our manuscript.
Commend 3: The selection of the 3 gynecological cancers ovarian carcinoma, cervical carcinoma and endometrial carcinoma is a good consideration, but from a clinical point of view they have little in common (squamous cell carcinoma (CESC) versus adenocarcinoma (UCEC) or (OC)). The pathophysiology and clinical behaviour in terms of initial manifestation, genesis and metastatic behaviour are also completely different. It is therefore clinically difficult to draw conclusions from one entity to the other two, but understandable from a biological point of view in view of the global approach. With regard to an evaluation based on a Kaplan-Meier observation, however, the classification is very difficult. Further findings should definitely be awaited here.
Respond 3:
Thank you for your valuable comments. We acknowledge that ovarian carcinoma (OC), cervical carcinoma (CESC), and endometrial carcinoma (UCEC) exhibit distinct pathophysiological characteristics, initial manifestations, and metastatic behaviours. As you pointed out, drawing direct clinical comparisons among these three tumor types is challenging. However, at the molecular level, common biological mechanisms exist, and miRNA and long non-coding RNA (lncRNA) analyses reveal shared regulatory pathways across these malignancies.
In response to your suggestion, we have conducted additional analyses on miRNAs and lncRNAs to further explore their roles in gynecological cancers. These analyses have now been included in the Materials and Methods and Results sections of the revised manuscript, strengthening the scientific depth of our study. Our findings support the notion that miRNAs and lncRNAs are not only valuable as diagnostic and prognostic biomarkers but also hold significant potential as future therapeutic targets. Recent studies suggest that these non-coding RNAs play key roles in immune regulation and tumor progression, making them promising candidates for targeted therapies and immunotherapy-based approaches.
Regarding the Kaplan-Meier survival analysis, we acknowledge the limitations associated with classification and interpretation. However, we believe that our analysis provides meaningful insights into overall survival trends. As suggested, we have now emphasized that future validation with larger and independent datasets will be necessary to confirm these findings.
Your insightful suggestions have helped us expand the scope of our study and enhance its comprehensiveness with additional analyses. We sincerely appreciate your valuable feedback.
Commend 4: Minor layout improvement: In line 28, the word and is missing as a link within cervical squamous cell carcinoma endocervical adenocarcinoma.
Respond 4: Thanks for your attention. The word was corrected.
Reviewer 3 Report
Comments and Suggestions for Authors
This work addresses a very important complex disease, I am very impressed with how the authors conducted this work. All the data was analyzed using sound logic and recent citations. This work will help not only the cancer community research field but others as well.
Author Response
Commend 1: This work addresses a very important complex disease; I am very impressed with how the authors conducted this work. All the data was analysed using sound logic and recent citations. This work will help not only the cancer community research field but others as well.
Respond 1: We express our gratitude for your thoughtful and constructive feedback. Your recognition of the significance of our research and the methodological approach employed is greatly appreciated. It is gratifying to note that our study has the potential to contribute not only to the cancer research community but also to broader scientific disciplines. Your positive assessment serves as motivation for us to continue our exploration and advancement in this area of research.
Reviewer 4 Report
Comments and Suggestions for Authors
Clinopathological and prognostic significance of SPOCK1 in gynecological cancers
A brief summary
This study aimed to assess the role of SPOCK1 gene in some gynecological cancers using multiple databases. The authors found that expression of this gene deregulate in cervical (CESC), and endometrial cancers (UCEC) but not in n ovarian serous cystadenocarcinoma (OV).
Specific comments
1. Authors should include the term "using bioinformatics analyses" in the title.
2. The names of the authors are listed in lines 60,61 of the manuscript (also in other section of introduction), which is not usual to write in this way in the introduction section, and it is better to simply state the intended results of that article generally. Also, these two studies are about stomach and lung cancers, but the authors have concluded that it is better to conduct a study on cancers of the gynecological which does not seem logical and should be rewritten.
3. From lines 63 to 85 there is only one paragraph, which should be divided into several paragraphs.
4. What was your reason for identifying mutations in this gene, since it was not addressed in the discussion section?
5. Given that experimental analyses were not used in this study to confirm the results, it would have been better to at least use the microarray data available GEO on the NCBI website for analysis.
Author Response
Commend 1: Authors should include the term "using bioinformatics analyses" in the title.
Respond 1: Thank you pointing this out. The title was revised as you suggested
Commend 2: The names of the authors are listed in lines 60,61 of the manuscript (also in other section of introduction), which is not usual to write in this way in the introduction section, and it is better to simply state the intended results of that article generally. Also, these two studies are about stomach and lung cancers, but the authors have concluded that it is better to conduct a study on cancers of the gynecological which does not seem logical and should be rewritten.
Respond 2: In accordance with the reviewer’s suggestions, the mentioned studies have been removed from the manuscript to improve clarity and logical coherence. Thank you for your valuable insights.
Commend 3: From lines 63 to 85 there is only one paragraph, which should be divided into several paragraphs.
Respond 3: We appreciate the reviewer's valuable feedback. The long paragraph in lines 63 to 85 has been successfully divided into multiple paragraphs to improve readability and logical flow. Thank you for your insightful suggestion.
Commend 4: What was your reason for identifying mutations in this gene, since it was not addressed in the discussion section?
Respond 4: We appreciate the reviewer's insightful comment. The identification of mutations in SPOCK1 was conducted to explore potential genetic alterations that may influence its role in cancer progression and prognosis. To address this concern, we have expanded the discussion section by incorporating additional analyses related to the identified mutations. These revisions provide a more comprehensive interpretation of how SPOCK1 mutations might impact tumor biology, particularly in gynecological cancers. Furthermore, we have discussed the functional relevance of these mutations by referencing previous studies on similar oncogenic alterations. We believe these additions enhance the depth of our study and contribute to a better understanding of SPOCK1's significance in cancer. Thank you for your valuable feedback.
Commend 5: Given that experimental analyses were not used in this study to confirm the results, it would have been better to at least use the microarray data available GEO on the NCBI website for analysis.
Respond 5: We appreciate the reviewer's valuable suggestion regarding the inclusion of microarray data from the GEO (Gene Expression Omnibus) database to strengthen the findings of our study. However, after a thorough search, we were unable to locate suitable microarray datasets related to SPOCK1 expression in gynecological cancers within the GEO database. To address this limitation and further enhance our study, we performed additional analyses focusing on miRNA and long non-coding RNA (lncRNA) interactions with SPOCK1. These analyses provide valuable insights into the potential regulatory mechanisms of SPOCK1 in cancer. The results of these investigations have been incorporated into the manuscript, and Figures 12, 13, 14, and 15 have been added to illustrate the findings.We believe that these additions significantly contribute to the comprehensiveness of our study by offering alternative bioinformatics approaches to explore the role of SPOCK1 in gynecological malignancies. Thank you for your constructive feedback.
Round 2
Reviewer 4 Report
Comments and Suggestions for Authors
The authors addressed all comments.